# Oligomerization of a molecular chaperone modulates its activity

**Tomohide Saio[1,2,3]\*, Soichiro Kawagoe[2], Koichiro Ishimori[1,2], Charalampos G Kalodimos[4]\***

[1]Department of Chemistry, Faculty of Science, Hokkaido University, Sapporo, Japan; [2]Graduate School of Chemical Sciences and Engineering, Hokkaido University, Sapporo, Japan; [3]PRESTO, Japan Science and Technology Agency, Kawaguchi, Japan; [4]Department of Structural Biology, St. Jude Children's Research Hospital, Memphis, United States

**Abstract** Molecular chaperones alter the folding properties of cellular proteins via mechanisms that are not well understood. Here, we show that Trigger Factor (TF), an ATP-independent chaperone, exerts strikingly contrasting effects on the folding of non-native proteins as it transitions between a monomeric and a dimeric state. We used NMR spectroscopy to determine the atomic resolution structure of the 100 kDa dimeric TF. The structural data show that some of the substrate-binding sites are buried in the dimeric interface, explaining the lower affinity for protein substrates of the dimeric compared to the monomeric TF. Surprisingly, the dimeric TF associates faster with proteins and it exhibits stronger anti-aggregation and holdase activity than the monomeric TF. The structural data show that the dimer assembles in a way that substrate-binding sites in the two subunits form a large contiguous surface inside a cavity, thus accounting for the observed accelerated association with unfolded proteins. Our results demonstrate how the activity of a chaperone can be modulated to provide distinct functional outcomes in the cell.
DOI: https://doi.org/10.7554/eLife.35731.001

**\*For correspondence:**
saio@sci.hokudai.ac.jp (TS);
babis.kalodimos@stjude.org (CGK)

**Competing interests:** The authors declare that no competing interests exist.

## Introduction

Molecular chaperones typically prevent the aggregation and assist with the folding of non-native proteins (*Balchin et al., 2016*; *Bukau et al., 2006*). Thus, chaperones are central to protein homeostasis in the cell and are essential for life (*Hipp et al., 2014*; *Powers and Balch, 2013*). Recent studies have also highlighted molecular chaperones as inhibitors of amyloid formation (*Mainz et al., 2015*; *Taylor et al., 2016*). Despite major advances in the field, how chaperones engage and alter the folding properties of non-native proteins remain poorly understood (*He et al., 2016*; *Huang et al., 2016*; *Koldewey et al., 2016*; *Libich et al., 2015*; *Rosenzweig et al., 2017*; *Saio et al., 2014*; *Sekhar et al., 2016*; *Verba et al., 2016*; *Wälti et al., 2017*). Despite common features, the mechanisms of activity are distinct in different families of chaperones (*Mattoo and Goloubinoff, 2014*). Studies of ATP-dependent chaperones, such as the Hsp70 and GroEL systems, have shown how cycles of ATP binding, hydrolysis and nucleotide release can give rise to different conformational states that exhibit distinct affinities for the substrate protein (*Apetri and Horwich, 2008*; *Clare et al., 2012*; *Hayer-Hartl et al., 2016*; *Kampinga and Craig, 2010*; *Mayer and Bukau, 2005*; *Saibil et al., 2013*; *Sekhar et al., 2016*; *Zhuravleva et al., 2012*). Much less is known about how ATP-independent chaperones assist with protein folding (*Stull et al., 2016*).

The trigger factor (TF) chaperone has several unique features (*Hoffmann et al., 2010*; *Ries et al., 2017*; *Wruck et al., 2018*): (i) is the only ribosome-associated chaperone in bacteria; (ii) with an estimated cellular concentration of ~50 µM (*Crooke et al., 1988*) it is also the most abundant one; (iii) in contrast to other oligomeric chaperones such as GroEL, SecB, and Hsp90 that form stable

oligomers, TF undergoes a dynamic transition between a monomeric and a dimeric form; (iv) TF functions both at the ribosome and in the cytosol: it binds, as a monomer, next to the exit channel at the ribosome to prevent the aggregation and premature folding of nascent polypeptides, while it functions as a dimer in the cytosol where is thought to assist in various processes in protein folding and biogenesis (*Agashe et al., 2004*; *Ferbitz et al., 2004*; *Haldar et al., 2017*; *Martinez-Hackert and Hendrickson, 2009*; *Oh et al., 2011*; *Ullers et al., 2007*). TF is also being widely used as a co-expression factor to improve the folding and yield of soluble proteins in biotechnology (*Uthailak et al., 2017*).

We recently determined the atomic resolution structure of TF in complex with a non-native protein (*Saio et al., 2014*). The structure revealed how the chaperone recognizes and engages the non-native protein and how it retains it in an unfolded state. Interestingly, our data showed that substrate protein binding causes TF to monomerize, thus indicating that the substrate-binding sites are occluded in dimeric TF. The interplay between substrate protein binding and chaperone oligomerization is likely to be used as a mechanism to modulate the energetics and kinetics of interaction in chaperone-substrate protein complexes, as for example in small heat shock proteins (*Eyles and Gierasch, 2010*). The large size of the dimeric TF (100 kDa) and its apparent dynamic nature has hindered determination of its structure.

We have taken advantage of recent advances in NMR spectroscopy and isotope labeling (*Huang and Kalodimos, 2017*) to determine the atomic structure of dimeric TF. The structure shows that three out of the five substrate-binding sites are partially buried in the dimer, thus explaining why protein binding results in TF monomerization. Interestingly, the dimer assembles in such a way that substrate-binding sites in the two subunits form a large contiguous surface inside a cavity. The structural data explain the unexpected finding that non-native proteins appear to bind with higher association rate to the dimeric TF than to the monomeric TF. Activity assays showed that TF dimerization enables the chaperone to exhibit stronger holdase and anti-aggregation activity.

## Results

### Characterization of TF dimerization

*Escherichia coli* TF consists of 432 amino acids, comprising RBD (residues 1 to 112), PPD (residues 150 to 246), and SBD (residues 113 to 149 and 247 to 432) (*Figure 1A*). Both multi-angle light scattering (MALS) and NMR studies show that TF forms

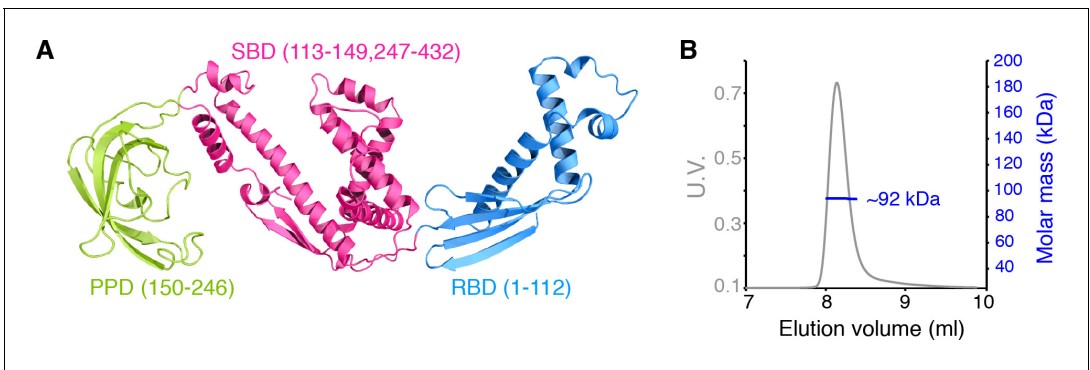

**Figure 1.** Dimerization of TF in solution. (**A**) Structure of *E. coli* TF (PDB code: 1W26). PPD, SBD, and RBD are shown in green, pink, and blue, respectively. The residue boundaries for each one of the three domains are shown in parentheses. SBD is discontinuous and is formed primarily by the C-terminal domain. (**B**) Size exclusion chromatography (SEC)-MALS of unliganded TF shows that the protein forms a dimer (Theoretical molar mass: 96 kDa) in solution.

DOI: https://doi.org/10.7554/eLife.35731.002

The following figure supplement is available for figure 1:

**Figure supplement 1.** Dynamic monomer-dimer transition of TF.

DOI: https://doi.org/10.7554/eLife.35731.003

a dimer of ~ 100 kDa in solution (*Figure 1B* and *Figure 1—figure supplement 1A*). MALS and analytical ultracentrifugation (AUC) experiments yielded a dimerization dissociation constant ($K_d$) of ~ 2 µM (*Figure 1—figure supplement 1B–D*), which is similar to previously reported values (*Kaiser et al., 2006*; *Maier et al., 2003*; *Morgado et al., 2017*). TF monomerization induced by substrate binding was previously reported (*Saio et al., 2014*) and has been further corroborated in the present work by MALS and NMR (*Figure 1—figure supplement 1A,E and F*). The intrinsic dissociation rate ($k_{diss}$) of the dimer was measured by tryptophan fluorescence following rapid dilution of TF (*Figure 1—figure supplement 1G*). Fitting of the data to a single exponential function resulted in $k_{diss}$ of ~10 s$^{-1}$, which indicates that the dimer is quite dynamic and thus the exchange between formation and dissociation of the dimer can be a major cause of the line broadening observed for the resonances located in RBD (*Morgado et al., 2017*) (*Saio et al., 2014*). This is further supported by the observation that line broadening at the interface of dimeric TF is suppressed in concentrated TF samples (~1 mM; *Figure 1—figure supplement 1A*). Previous studies employing fluorescent labeling (*Kaiser et al., 2006*) or (1-oxyl-2,2,5,5-tetramethyl-3-pyrroline-3-methyl)-ethanethiosulfonate (MTSL) spin labeling tags (*Morgado et al., 2017*) reported slower dissociation rates, likely due to the strong hydrophobic nature of the tag.

Structure of dimeric TF. We used NMR spectroscopy to determine the structure of the 100 kDa dimeric form of TF in solution (see Materials and methods). We used U-$^{12}$C,$^{15}$N-labelled TF samples that contained specifically protonated methyl groups of Ala, Val, Leu, Met, Thr and Ile ($\delta$1) and protonated aromatic residues Phe, and Tyr in an otherwise deuterated background (*Huang and Kalodimos, 2017*; *Tzeng et al., 2012*) (*Figure 2—figure supplement 1*). The high sensitivity and resolution of the methyl region, combined with the high abundance of these eight amino acids in TF (*Figure 2—figure supplement 1*) and in the dimeric TF interface provided a large number of intra- and inter-molecular nuclear Overhauser effects (NOEs) (*Table 1*).

The structure of dimeric TF is shown in *Figure 2*. TF forms a symmetric dimer in a head-to-tail orientation. Part of RBD inserts into a large cavity that is formed between the SBD and PPD of the other subunit (*Figure 2A and B*) and the arrangement results in three major interfaces that hold the dimer together (*Figure 2C*). The three helices in RBD ($\alpha$1-$\alpha$3) form extensive contacts with PPD and the SBD arm 1 and arm 2 regions. Specifically, a hydrophobic patch in SBD arm 1 consisting of bulky hydrophobic residues (Leu314, Phe322, Leu332, Leu336, and Phe337) forms intimate nonpolar contacts with the C-terminal region of RBD helix $\alpha$1 (Val35, Ala36, Val39, and Ile41) (*Figure 2C*). This binding interface is further strengthened by a salt bridge between Arg40 and Glu339 and a hydrogen bond between Lys38 and Gln340. SBD arm 1 also interacts with the N-terminal part of RBD helix $\alpha$3 exclusively via polar contacts (e.g. between Asp65 and Arg321) (*Figure 2C*). 2,540 Å$^2$ (1,620 Å$^2$ nonpolar and 920 Å$^2$ polar) of surface are buried in this interface of the dimer. A large hydrophobic patch in PPD consisting of aromatic and bulky nonpolar residues (Phe168, Phe185, Met194, Ile195, Tyr221) engages the long loop in RBD connecting helices $\alpha$1 and $\alpha$3, which also features a short helix ($\alpha$2). Residues Phe44, Val49, Ile53, and Tyr58 in RBD appear to establish the most important contacts with PPD, including two salt bridges (between Arg57 and Asp184 and between Lys48 and Glu199) (*Figure 2C*). 2,650 Å$^2$ (1,900 Å$^2$ nonpolar and 750 Å$^2$ polar) of surface are buried in this interface. The third major dimeric interface is mediated by SBD arm 2 and the C-terminal region of RBD helix $\alpha$3. Similar to the other two, this interface is made up primarily of nonpolar residues (RBD residues Ile76, Ile79, Ile80 and Ile84; SBD residues Val384, Tyr388, Phe387, and Leu394) with additional salt bridges at the periphery of the binding site. 1,480 Å$^2$ (1,050 Å$^2$ nonpolar and 430 Å$^2$ polar) are buried at this dimeric interface, which is the smallest among the three ones. A total surface of 6,670 Å$^2$ is buried upon dimer formation. The extensive interface seen in the structure to mediate the dimer was tested by mutagenesis and a triple amino-acid substitution variant (V39E/I76E/I80A; hereafter TF$^{mon}$) was identified that abolishes TF dimerization (*Figure 2—figure supplement 2*). A recently reported low-resolution structural model of TF dimer also showed a head-to-tail orientation of the two subunits (*Morgado et al., 2017*). However, the dimeric interface is very different from the one observed in our structure (*Figure 2—figure supplement 3*).

Superposition of the crystallographically determined structure of the monomeric TF (*Ferbitz et al., 2004*) on one of the subunits of the dimeric TF demonstrates that TF undergoes major conformational changes as it transitions from the monomeric to the dimeric state (*Figure 3A*). With respect to SBD, RBD undergoes a 60° rotation and a 21 Å translation, whereas PPD undergoes a 74° rotation and a 19 Å translation. As a result, PPD moves closer to SBD within the same subunit,

**Table 1.** Structural and NMR statistics of TF dimer.

**Distance restraints***

| | | |
|---|---|---|
| NOEs | | |
| | Short range (intraresidue and sequential) | 870 |
| | Medium range (2 < \| i-j \| < 5) | 467 |
| | Long range ( \| i-j \| > 5 ) | 1230 |
| | Intermolecular | 54 |
| Hydrogen bonds | | 374 |
| Dihedral angle restraints (cp and | | 1358 |
| Violations (mean and SD)* | | |
| | Distance restraints (A) | 0.005 ± 0.025 |
| | Dihedral angle restraints (°) | 0.02 ± |
| 0.23 | | |
| Structural coordinates rmsd* | | |
| | RBD core (1-39, 51-112) | |
| | Chain A | |
| | Backbone atoms | 1.50 ± 0. A |
| | All heavy atoms | 2.04 ± 0.29 A |
| | Chain B | |
| | Backbone atoms | 1.56 ± 0.41 A |
| | All heavy atoms | 2.07 ± 0.38 A |
| | PPD core (157-190,195-241) | |
| | Chain A | |
| | Backbone atoms | 0.87 ± 0.09 A |
| | All heavy atoms | 1.38 ± 0.07 A |
| | Chain B | |
| | Backbone atoms | 0.82 ± 0.14 A |
| | All heavy atoms | 1.30 ± 0.11 A |
| | SBD core (115-149, 250-321, 329-428) | |
| | Chain A | |
| | Backbone atoms | 1.40 ± 0.21A |
| | All heavy atoms | 2.17 ± 0.23 A |
| | Chain B | |
| | Backbone atoms | 1.34 ± 0.16A |
| | All heavy atoms | 2.14 ± 0.20 A |
| Ramachandran plot* | | |
| | Most-favored regions | 85.4% |
| | Additionally allowed regions | 14.3% |
| | Generously allowed regions | 0.3% |
| | Disallowed regions | 0.0% |

*The statistics apply to the 20 lowest-energy structures.

DOI: https://doi.org/10.7554/eLife.35731.004

and the two domains form a large cavity wherein the RBD of the other subunit inserts into (*Figures 2* and *3A*). These conformational changes results in a more compact TF structure in the dimeric form, which is consistent with small-angle X-ray scattering (SAXS) data (*Ries et al., 2017*).

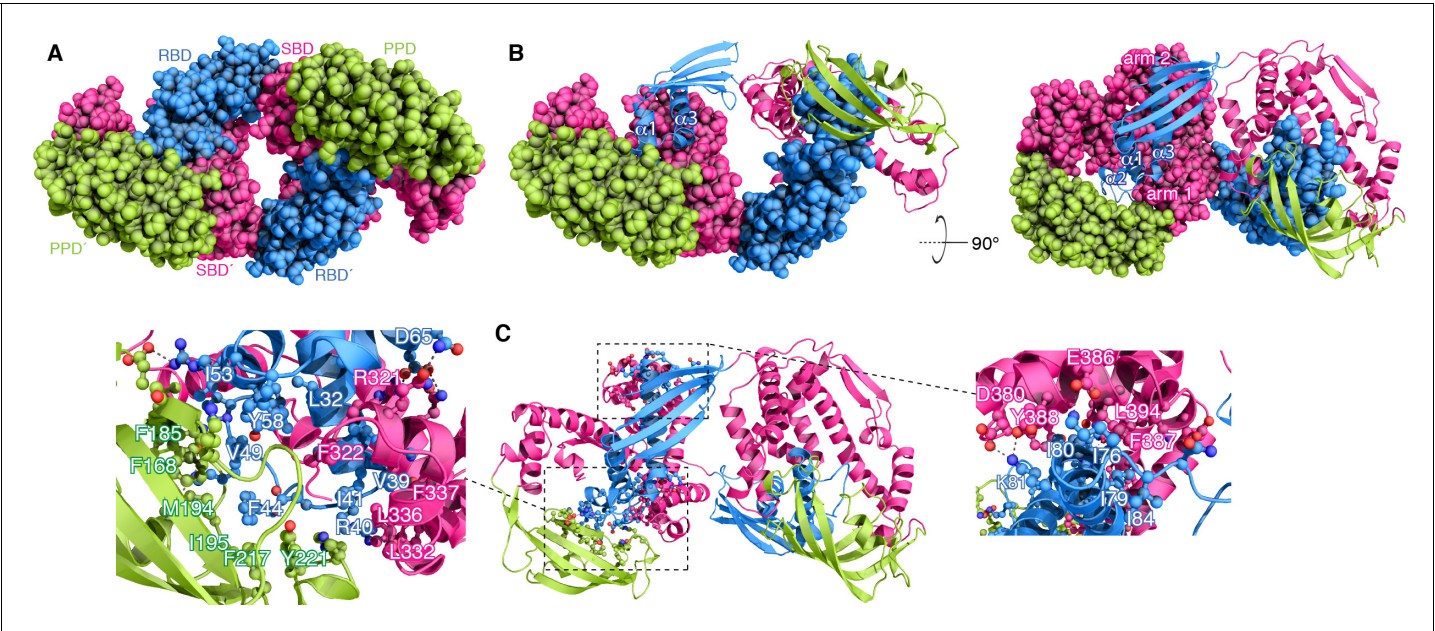

**Figure 2.** Structural basis for TF dimerization. (**A**) The lowest-energy structure of the TF dimer is shown as space-filling model. TF forms a dimer in a head-to-tail orientation. RBD, SBD, and PPD are shown in blue, magenta, and green, respectively. (**B**) One of the TF subunits is shown as space-filling model and the other subunit shown in ribbon. The helices of the RBD and the two arm regions are labeled. (**C**) Expanded views of the dimeric interfaces highlighting contacts between the two subunits. Residues involved in mediating dimerization are shown as ball-and-stick.

DOI: https://doi.org/10.7554/eLife.35731.005

The following figure supplements are available for figure 2:

**Figure supplement 1.** NMR of dimeric TF.
DOI: https://doi.org/10.7554/eLife.35731.006

**Figure supplement 2.** SEC-MALS of TF mutants.
DOI: https://doi.org/10.7554/eLife.35731.007

**Figure supplement 3.** Comparison with PRE-based docking models.
DOI: https://doi.org/10.7554/eLife.35731.008

**Figure supplement 4.** Examples of the inter-molecular NOEs.
DOI: https://doi.org/10.7554/eLife.35731.009

Dimerization buries the ribosome- and substrate-binding sites. The structural data suggest that TF dimerization has profound impact on the function of TF because the ribosome-binding region as well as several of the substrate-binding sites are buried in the dimer (*Figure 3B* and *Figure 3—figure supplement 1*). The RBD loop, which contains the signature motif ($G^{43}FRxGxxP^{50}$) mediating the interaction of TF with the ribosome, is sequestered by the PPD of the other subunit in the TF dimer and thus is not available for binding to the ribosome. This finding explains why TF must monomerize upon binding to the ribosome (*Ferbitz et al., 2004*). The intrinsic affinity of TF for the ribosome (*Kaiser et al., 2006*; *Maier et al., 2003*) ($K_d$ ~0.5 µM; *Figure 3—figure supplement 1C*) is comparable to the dimerization $K_d$ (~2 µM); therefore, there is a strong competition between TF dimerization and ribosome binding. Because the affinity of TF for ribosome-nascent-chain (RNC) complexes is substantially stronger ($K_d$ <0.01 µM) (*Bornemann et al., 2014*; *Rutkowska et al., 2008*) than for vacant ribosomes, translating ribosomes will be invariably bound, and thus protected by TF.

TF uses five distinct binding sites (*Figure 3B*) to interact with unfolded substrates such as the maltose binding protein (MBP) (*Saio et al., 2014*). Four of these substrate-binding sites are located in SBD (A-D) and the fifth one (E) is located in PPD. In the dimeric form of TF only two (A and D) among these five substrate-binding sites are fully accessible for binding, whereas the other three (B, C, and E) are partially occluded (*Figure 3B*). A protein substrate typically engages at least four of the binding sites (*Saio et al., 2014*); thus, complex formation between TF and an unfolded protein requires that TF monomerize, as supported by NMR and MALS data (*Figure 1—figure supplement*

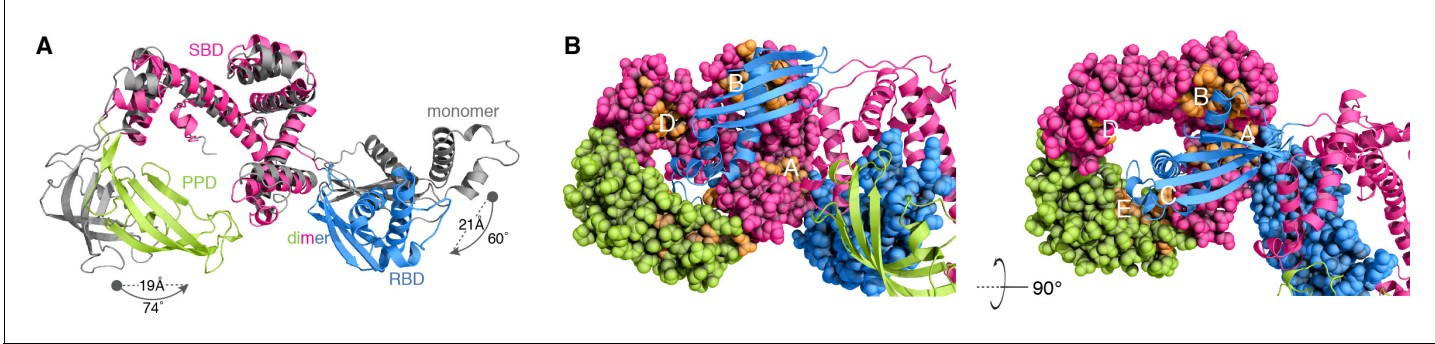

**Figure 3.** Conformational changes of TF upon dimerization. (**A**) The structure of one subunit in the TF dimer (colored as in *Figure 1A*) and the crystal structure of monomeric TF (colored grey) [Protein Data Bank (PDB) code: 1W26] are superimposed for SBD. The changes in rotation and translation of the RBD and PPD between the monomer and the dimer are indicated. (**B**) View of the structure of dimeric TF highlighting the positioning of the substrate-binding sites (colored orange). The five main substrate-binding sites are labeled A, B, C, D, and E.

DOI: https://doi.org/10.7554/eLife.35731.010

The following figure supplements are available for figure 3:

**Figure supplement 1.** The ribosome-binding loop in the TF dimer.

DOI: https://doi.org/10.7554/eLife.35731.011

**Figure supplement 2.** Characterization of small substrate proteins in complex with TF.

DOI: https://doi.org/10.7554/eLife.35731.012

*1A,E and F*) (*Saio et al., 2014*). Previous crystallographic data indicated that TF may also bind to small folded proteins as a dimer (*Martinez-Hackert and Hendrickson, 2007*). However, NMR characterization of such complexes in solution showed that the substrates are in an unfolded state and TF is in the monomeric state (*Figure 3—figure supplement 2*).

Dimerization modulates the chaperone activities of TF. The concentration of TF in the cell (~50 μM) is 2 to 3-fold that of the ribosome (*Patzelt et al., 2002*), and given the low $K_d$ of dimerization (~2 μM) the vast majority of free TF in the cytoplasm will exist in the dimeric form. The dissociation rate ($k_{diss}$) of the dimer is ~ 10 $s^{-1}$ indicating a rather dynamic TF dimer with a residence time of ~ 100 ms (*Figure 1—figure supplement 1G*). We sought to investigate whether the chaperone activity of TF is affected as it transitions between the monomeric and dimeric forms. To characterize the chaperone activity of the monomeric form of TF, we used the $TF^{mon}$ variant (*Figure 2—figure supplement 2*). The amino acid substitutions that abolish dimerization in this variant are located in RBD and thus do not affect protein substrate binding. First, we performed aggregation assay using the 35 kDa protein glyceraldehyde-3-phosphate dehydrogenase (GAPDH) in the absence and presence of TF or $TF^{mon}$. Denatured GAPDH was diluted into buffer and its aggregation was monitored by light scattering. The results showed that dimeric TF was substantially more efficient at suppressing aggregation than the monomeric TF (*Figure 4A* and *Figure 4—figure supplement 1*). Interestingly, decreased anti-aggregation activity of another monomeric variant, $TF^{\Delta RBD}$ (*Figure 2—figure supplement 2E*), has also been reported previously (*Merz et al., 2006*). Note that the results for $TF^{\Delta RBD}$ and TF monomeric mutant are essentially identical in our GAPDH aggregation and MBP refolding assays. The anti-aggregation assay was also performed using a shorter substrate protein $OmpA^{1-192}$ (*Figure 4B*). The results showed that in the case of shorter substrate, which has a smaller number of hydrophobic regions (*Figure 4—figure supplement 2*), the difference in the anti-aggregation activity between the dimeric TF and monomeric TF is much less pronounced and both species are equally efficient in suppressing aggregation.

Next, we examined the efficiency of TF in assisting with the folding of MBP. Denatured MBP was diluted into buffer and its refolding was monitored by the characteristic increase in tryptophan fluorescence intensity in the absence and presence of TF or $TF^{mon}$ (*Figure 4C*) (*Apetri and Horwich, 2008*; *Chakraborty et al., 2010*). At 1:1 stoichiometric ratio with MBP, $TF^{mon}$ had a minimal effect on MBP folding whereas dimeric TF had a pronounced effect (*Figure 4C*). Specifically, dimeric TF increased the apparent folding rate of MBP and at the same time increased the yield of the soluble fraction substantially (*Figure 4C and D*). The increase in the apparent folding rate is likely due to the

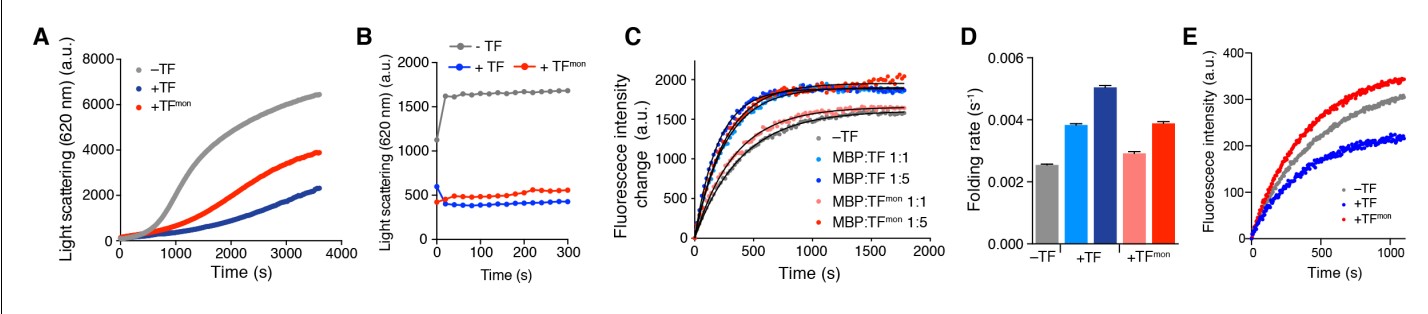

**Figure 4.** Effect of TF dimerization on chaperone activities. Aggregation of GAPDH in the absence or presence of TF and TF$^{mon}$ at 0.5 µM (**A**) and OmpA in the absence or presence of TF and TF$^{mon}$ at 4 µM (**B**). (**C**) Refolding of MBP in the absence or presence of TF and TF$^{mon}$. The solid line represents the fit of the data to a single exponential function. (**D**) Folding rates of MBP from the analysis of the curves shown in panel C. (**E**) Refolding of the slowly-folding MBPY283D variant in the absence or presence of TF and TF$^{mon}$.
DOI: https://doi.org/10.7554/eLife.35731.013

The following figure supplements are available for figure 4:

**Figure supplement 1.** Aggregation of GAPDH in the absence or presence of 1 µM TF and TF$^{mon}$.
DOI: https://doi.org/10.7554/eLife.35731.014

**Figure supplement 2.** Sequence hydrophobicity of the substrate proteins of TF (Roseman algorithm, window = 9).
DOI: https://doi.org/10.7554/eLife.35731.015

most efficient suppression of aggregation by the dimeric TF (*Apetri and Horwich, 2008*). Notably, a much higher TF$^{mon}$ concentration was needed to match the chaperone activity of the dimeric TF (*Figure 4C and D*). We also tested the effect of TF on an aggregation-prone, slowly folding mutant of MBP (MBP$^{Y283D}$) (*Huang et al., 2016*; *Saio et al., 2014*). The dimeric TF was observed to have a strong 'holdase' effect on the mutant MBP as evidenced by the suppression of the folding of MBP$^{Y283D}$ (*Figure 4E*). In contrast, TF$^{mon}$ slightly accelerated folding (*Figure 4E*). Because refolding of MBP$^{Y283D}$ was performed in a chloride-free buffer in which MBP does not aggregate (*Apetri and Horwich, 2008*), any contribution of an anti-aggregation effect can be excluded. Taken together, all assays showed that the monomeric and dimeric TF states have distinct chaperone activities.

TF dimerization accelerates its association rate with substrates. To understand how the oligomeric state of TF affects chaperone activity, we sought to determine how the monomeric and dimeric TF species interact with protein substrates. ITC showed that TF$^{mon}$ has a 5-fold higher affinity ($K_d \sim 6$ µM) for protein substrates than the dimeric TF ($K_d \sim 35$ µM) (*Figure 5—figure supplement 1*). This is expected given that a sizable fraction of the substrate-binding surface is buried in the dimeric TF (*Figure 3B* and *Figure 5—figure supplement 2A*). Next, we measured the kinetics of substrate binding to TF using stopped-flow fluorescence spectroscopy. Notably, the rates of protein substrate association and dissociation are very different for the dimeric (*Figure 5A–C*) and monomeric TF (*Figure 5D and E*). Specifically, unfolded PhoA binds TF$^{mon}$ with a $k_{on} \sim 0.5 \times 10^6$ M$^{-1}$ s$^{-1}$ and dissociates with a $k_{off} \sim 6$ s$^{-1}$. In comparison, dimeric TF binds PhoA with a 2-fold faster association rate ($k_{on} \sim 1.1 \times 10^6$ M$^{-1}$ s$^{-1}$) and dissociates with a 5-fold faster dissociation rate ($k_{off} \sim 30$ s$^{-1}$). The faster association of non-native proteins with the dimeric TF over the monomeric TF is consistent with the stronger holdase activity of dimeric TF (*Figure 4E*). Note that the holdase activity of a chaperone is determined by the difference between the folding rate of the substrate protein and the association rate between the chaperone and the unfolded substrate protein, as shown by kinetic experiments on SecB and TF (*Huang et al., 2016*). Thus the association rate of dimeric TF for the substrate appears to be fast enough to delay the folding of the slowly folding mutant MBP$^{Y283D}$, but not fast enough to delay the folding of wild type MBP (*Figure 4C and E*). Although three out of the five substrate-binding sites are partially occluded in the dimeric TF, assembly of the dimer brings next to each other substrate-binding sites A, B, and D in the two subunits. The sites are located within a large cavity that is accessible to unfolded proteins (*Figure 5—figure supplement 2*) and present to the substrate a large contiguous binding surface that may account for the enhanced association rates of substrates with the dimeric TF.

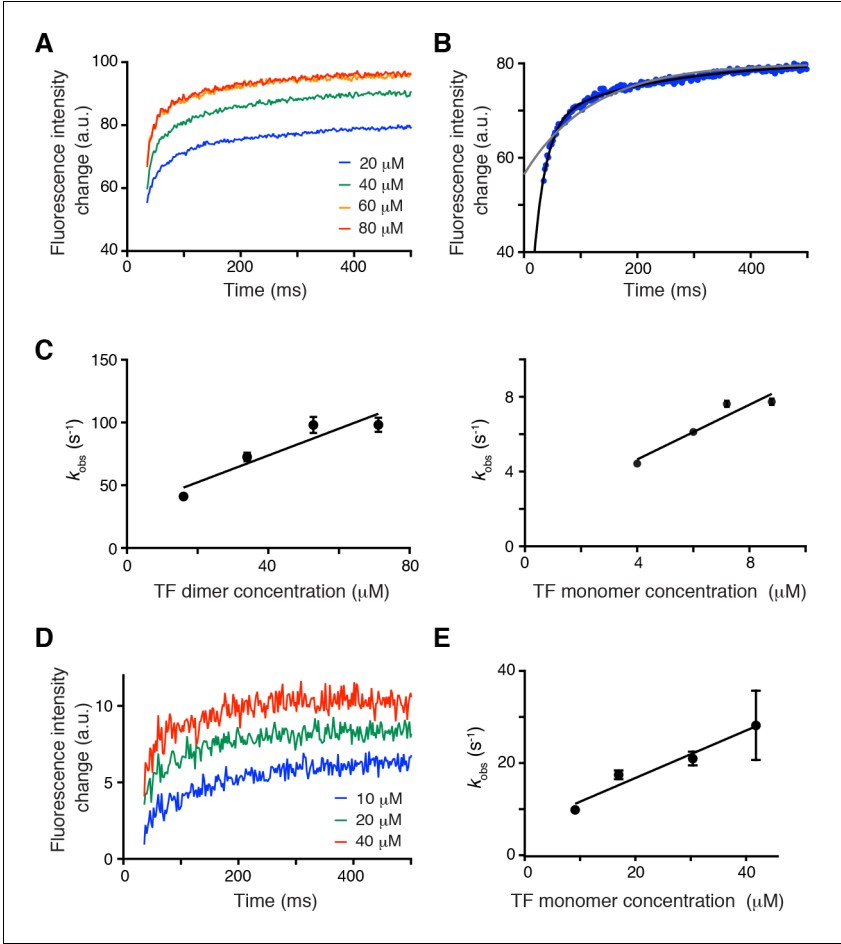

**Figure 5.** Effect of TF dimerization on binding kinetics. (**A**) Association of unfolded PhoA with TF monitored by tryptophan fluorescence. (**B**) Fitting of the data for the association of PhoA with TF by a single exponential function (gray line) or the sum of two exponential functions (black line), indicating that two exponential functions are required to fit the data. (**C**) Plots of the observed rate constant ($k_{obs}$) as a function of the concentration of the dimer (left) and the monomer (right) of TF. (**D**) Association of PhoA with TFmon monitored by tryptophan fluorescence. (**E**) Plot of the observed rate constant ($k_{obs}$) as a function of the concentration of monomeric TF.
DOI: https://doi.org/10.7554/eLife.35731.016

The following figure supplements are available for figure 5:

**Figure supplement 1.** ITC traces of the titration of unfolded proteins to TF.
DOI: https://doi.org/10.7554/eLife.35731.017
**Figure supplement 2.** Substrate-binding sites in the dimer.
DOI: https://doi.org/10.7554/eLife.35731.018

## Discussion

Our findings demonstrate how changes in the oligomerization state of a molecular chaperone may modulate the folding properties to interacting substrate proteins. The structural, energetic and kinetic data presented here explain previous observations and offer new insights into the various roles of TF in the cytoplasm (*Figure 6*). When bound to the ribosome (*Figure 6*, panel i), TF is in the monomeric form and exposes all substrate-binding sites to the nascent protein. The co-localization with the nascent chain results in TF delaying folding and preventing aggregation as shown before (*Agashe et al., 2004*; *Hoffmann et al., 2012*; *O'Brien et al., 2012*; *Saio et al., 2014*). As the nascent chain grows, additional TF molecules are recruited (*Figure 6*, panel ii) (*Kaiser et al., 2006*). Because of the high concentration of free TF in the cytoplasm, it is likely that a TF molecule outcompetes and displaces the fraction of the nascent chain that is bound to the TF to form a TF dimer. In

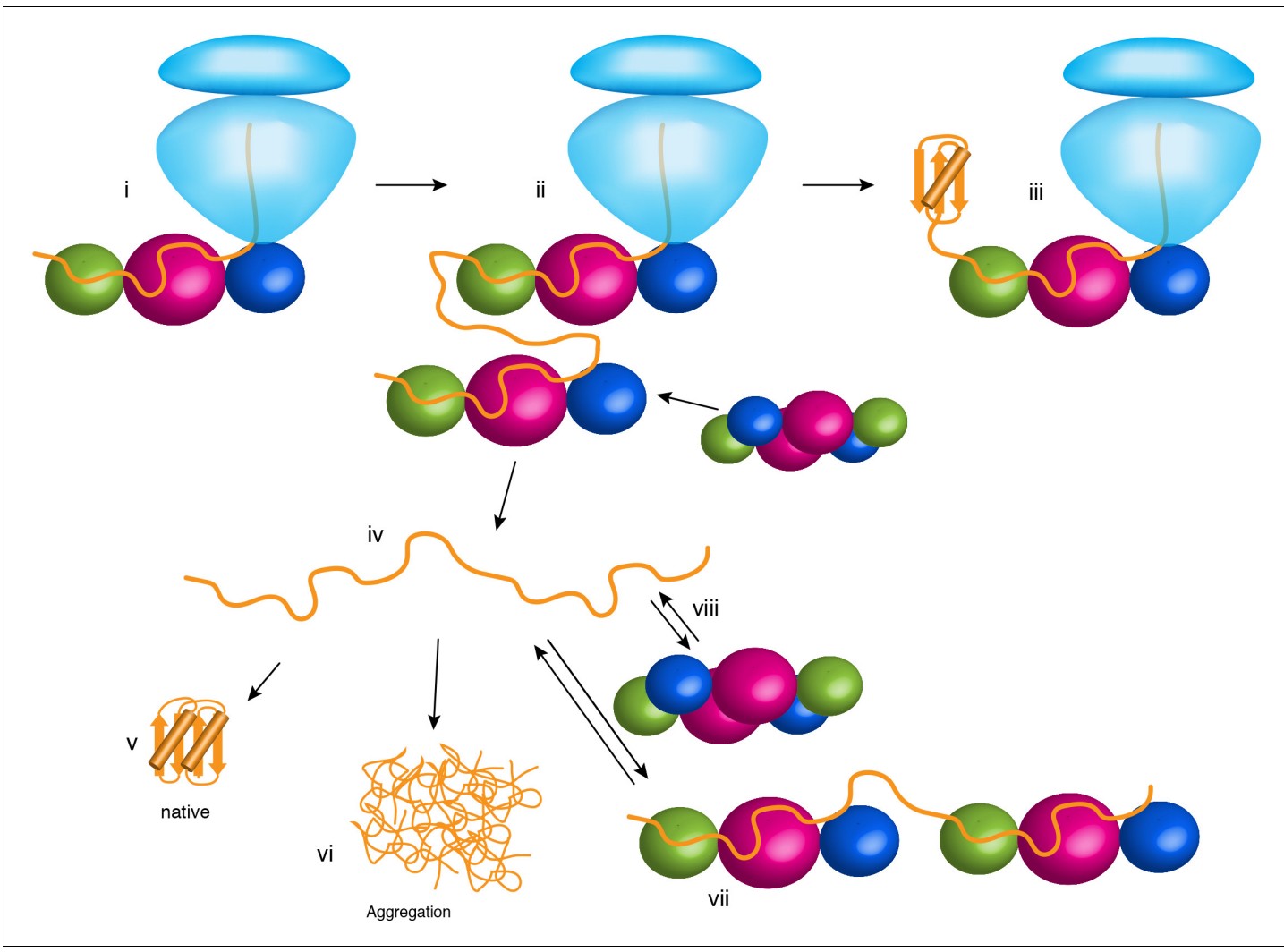

**Figure 6.** Chaperone activities of TF in the cell. The ribosome is shown in light blue. The protein substrate is shown in orange, and TF is represented as spheres with the subunits colored as in *Figure 2A*. See text for details.

DOI: https://doi.org/10.7554/eLife.35731.019

this case, folding of a domain may occur co-translationally (*Figure 6*, panel iii). Most cytosolic proteins released from the ribosome (*Figure 6*, panel iv) spontaneously form their native structure (*Balchin et al., 2016*) (*Figure 6*, panel v). However, in the absence of molecular chaperones, many proteins have a tendency to aggregate (*Figure 6*, panel vi). TF has an anti-aggregation activity, with the dimeric form being more potent than the monomeric form (*Figure 4A and C*). We posit that this is because of the higher local concentration of TF subunits in the dimeric form, which can both bind upon dimer dissociation with the interacting non-native polypeptide to protect longer segments of the polypeptide (*Figure 6*, panel vii). The increased local concentration of TF subunits results in faster association of the second molecule of TF to the substrate protein, which enables TF to more efficiently capture the substrate protein before the it starts to aggregate. This hypothesis is consistent with the following findings: (i) higher concentrations of monomeric TF are needed to achieve the same anti-aggregation activity as the dimeric TF (*Figure 4C*); and (ii) aggregation of shorter substrates, which are expected to bind to a single TF molecule, are equally prevented by the monomeric and dimeric TF (*Figure 4B*). Depending on the energetics and kinetics of interaction between TF and the non-native polypeptide in the cytoplasm, dimeric TF can also function as a potent holdase chaperone (*Figure 4E*) to delay the folding of proteins destined for export, such as periplasmic and outer membrane proteins (*Oh et al., 2011*). Our findings demonstrate how the activity of a

chaperone can be modulated and tailored to specialized needs in the cell simply by a change in the oligomeric state of the chaperone, without the need of ATP binding and hydrolysis cycles or the binding of co-factors.

# Materials and methods

## Key resources table

| Reagent type (species) or resource | Designation | Source or reference | Identifiers |
|---|---|---|---|
| Strain, strain background (*E. coli*) | BL21 (DE3) | NIPPON GENE CO., LTD. | ECOS Competent E. coli BL21 (DE3) |
| Recombinant DNA reagent | PhoA | *Saio et al. (2014)*, PMID: 24812405 | NCBIGene:945041 |
| Recombinant DNA reagent | OmpA | *Tsirigotaki et al., (2018)*, PMID: 29606594 | NCBIGene:945571 |
| Recombinant DNA reagent | RT | *Inouye et al., (1999)*, PMID: 10531319 | UniProtKB: P23070 |
| Recombinant DNA reagent | MBP | *Huang et al. (2016)*, PMID: 27501151 | NCBIGene: 948538 |
| Recombinant DNA reagent | TF | Takara Bio inc. | pCold-TF (TKR 3365) |
| S7 | S7 | GenScript | Gene synthesis |
| Peptide, recombinant protein | GAPDH | Sigma-Aldrich | G-2267 |
| Software, algorithm | CYANA3.97 | *Güntert (2004)*, PMID: 15318003 | RRID:SCR_014229 |
| Software, algorithm | CNS | *Brunger (2007)*, PMID: 18007608 | RRID:SCR_014223 |

## Sample preparation

The *E. coli* TF, RBD (residues 1 to 117), PPD (residues 148 to 249), SBD (residues 113–432Δ150–246), TF$^{\Delta RBD}$ (residues 113 to 246), TF$^{\Delta PPD}$ (residues 1–432Δ150–243), OmpA$^{1-192}$, and PhoA were expressed and purified as described previously (*Saio et al., 2014*). TF, PPD, TF$^{\Delta RBD}$, and TF$^{\Delta PPD}$ were cloned into the pCold vector (Takara Bio). RBD and SBD were cloned into pET16b vector (Novagen) and fused to His$_6$-MBP and a tobacco etch virus (TEV) protease cleavage site. TF mutants were constructed by site-directed mutagenesis using PfuTurbo High Fidelity DNA polymerase (Agilent) as well as PrimeSTAR Max (Takara Bio). OmpA$^{1-192}$ was fused with N-terminal His$_6$-tag and cloned into pET16b. Precursor form of maltose-binding protein (preMBP) and MBP$^{Y283D}$ were expressed and purified as described previously (*Huang et al., 2016*). *E. coli* reverse transcriptase (RT)-Ec86 255–320 was cloned into pCold-TF (Takara Bio) including a ~ 25 a.a. linker between TF and RT. *E. coli* S7 was cloned into pET16b vector and fused to His$_6$-MBP, including a tobacco etch virus (TEV) protease cleavage site. All constructs were transformed into *E. coli* BL21 (DE3) cells.

For the unlabeled protein samples, cells were grown in Luria-Bertani (LB) medium at 37°C in the presence of ampicillin (100 mg L$^{-1}$). Protein expression was induced by the addition of 0.2 to 0.5 mM isopropyl-β-D-1-thiogalactopyranoside (IPTG) at OD$_{600}$ ~ 0.6, followed by ~16 hr of incubation at 18°C. For isotopically labeled samples for NMR studies, cells were grown in minimal (M9) medium at 37°C in the presence of ampicillin (100 mg L$^{-1}$). Protein expression was induced by the addition of 0.2 to 0.5 mM IPTG at OD$_{600}$ ~ 0.6, followed by ~16 hr of incubation at 18°C. The samples with $^1$H,$^{13}$C-labeled methyl and aromatic side chains in deuterium background were prepared as described previously (*Saio et al., 2014*). The cells were grown in medium with $^{15}$NH$_4$Cl (1 gL$^{-1}$) and $^2$H$_7$-glucose (2 gL$^{-1}$) in 99.9% $^2$H$_2$O (CIL and Isotec). For preparation of $^1$H-$^{13}$C methyl-labeled samples, α-ketobutyric acid (50 mg L$^{-1}$) and α-ketoisovaleric acid (85 mg L$^{-1}$), [$^{13}$CH$_3$] methionine (50 mg L$^{-1}$), [$^2$H$_2$, $^{13}$CH$_3$] alanine (50 mg L$^{-1}$) were added to the culture 1 hr before the addition of IPTG. For Phe and Tyr labeling, U-[$^1$H, $^{13}$C]-labeled amino acids were added to the culture 1 hr before the addition of IPTG.

Cells were harvested and resuspended in the lysis buffer containing 50 mM Tris-HCl pH 8.0, 500 mM NaCl. Cells were disrupted by a high-pressure homogenizer or sonicator and centrifuged at 50,000 g for 45 min. TF, TF variants, and PhoA fragments were purified using Ni Sepharose 6 Fast Flow resin (GE Healthcare). In the case of RBD, SBD, and PhoA fragments that contain TEV cleavage site, the His$_6$-MBP tag was removed by TEV protease at 4°C (incubation for 16 hr). The proteins were further purified by gel filtration using Superdex 75 16/60 or 200 16/60 columns (GE Healthcare). TF-RT complex was purified using Ni Sepharose 6 Fast Flow resin, followed by gel filtration using Superdex 200 16/60 column equilibrated with a solution containing 20 mM potassium phosphate (pH 7.0), 100 mM KCl, 4 mM β-mercaptoethanol, 0.5 mM EDTA, 0.05% NaN$_3$. S7 was purified using Ni Sepharose 6 Fast Flow resin, followed by the removal of His$_6$-MBP tag by TEV protease digestion at 4°C in the presence of TF. TF-S7 complex was further purified by gel filtration using Superdex 200 16/60 column equilibrated with a solution containing 20 mM potassium phosphate (pH 7.0), 100 mM KCl, 4 mM β-mercaptoethanol, 0.5 mM EDTA, 0.05% NaN$_3$. MBP$^{Y283D}$ and preMBP was purified using Ni Sepharose 6 Fast Flow resin, followed by gel filtration using Superdex 200 16/60 column equilibrated with a solution containing 100 mM HEPES, pH 7.5, 20 mM potassium acetate, 5 mM magnesium acetate. For OmpA$^{1-192}$, the cell pellet was resuspended in a solution containing 50 mM Tris-HCl (pH 8.0), 500 mM NaCl, and 8 M urea and incubated for 1 hr at room temperature, followed by centrifugation at 50,000 g for 45 min. The solubilized protein was purified using Ni Sepharose 6 Fast Flow resin, and eluted with a solution containing 50 mM Tris-HCl (pH 8.0), 500 mM NaCl, 400 mM imidazole, and 8 M urea.

## NMR spectroscopy

For NMR titrations and NOE measurement, NMR samples were prepared in 20 mM potassium phosphate (pH 7.0), 100 mM KCl, 4 mM β-mercaptoethanol, 0.5 mM EDTA, 0.05% NaN$_3$, and 7% $^2$H$_2$O. The proteins were concentrated to 0.3 ∼ 2.2 mM for NOESY measurements. NMR spectra were recorded on Agilent UNITY Inova 600 and 800 MHz NMR spectrometers and Bruker Avance III 600, 700, and 800 MHz NMR spectrometers. Bruker Avance III 700 was equipped with cryogenic probe. The experiments were run at 10, 22, and 35°C. Spectra were processed using the NMRPipe program (*Delaglio et al., 1995*), and data analysis was performed with Olivia (fermi.pharm.hokudai.ac.jp/olivia). NOE distance restraints for the dimer was collected by $^{13}$C-edited NOESY-HMQC, 3D ($^1$H)-$^{13}$C HMQC-NOESY-$^1$H-$^{13}$C HMQC, 3D-SOFAST-($^1$H)-$^{13}$C HMQC-NOESY-$^1$H-$^{13}$C HMQC and $^{13}$C-edited SOFAST-NOESY-HMQC (*Rossi et al., 2016*) recorded on [U-$^2$H; Ala-$^{13}$CH$_3$; Met-$^{13}$CH$_3$; Ile-δ1-$^{13}$CH$_3$; Leu/Val-$^{13}$CH$_3$/$^{13}$CH$_3$; Phe-$^{13}$C$^{15}$N; Tyr-$^{13}$C$^{15}$N]-labeled TF or on 1:1 mixture of [U-$^2$H; Ala-$^{13}$CH$_3$; Met-$^{13}$CH$_3$; Ile-δ1-$^{13}$CH$_3$]-labeled TF and [Leu/Val-$^{13}$CH$_3$/$^{13}$CH$_3$; Phe-$^{13}$C$^{15}$N; Tyr-$^{13}$C$^{15}$N]-labeled TF. The 1:1 mixture of the TF proteins with different labeling schemes enabled us to unambiguously identify the intermolecular NOEs: For example, an NOE observed between Ile and Leu can be unambiguously classified as an intermolecular NOE. Although the resonances from the interface, especially from RBD, undergo severe line broadening, high sensitivity of methyl resonances in deuterated background as well as high solubility and stability of TF at wide range of temperature enabled observation of substantial number of NOEs. NOEs were further collected by 3D ($^1$H)-$^{13}$C HMQC-NOESY-$^1$H-$^{13}$C HMQC, 3D-SOFAST-($^1$H)-$^{13}$C HMQC-NOESY-$^1$H-$^{13}$C HMQC, and $^{13}$C-edited SOFAST-NOESY-HMQC recorded on [U-$^2$H; Ala-$^{13}$CH$_3$; Met-$^{13}$CH$_3$; Ile-δ1-$^{13}$CH$_3$; Leu/Val-$^{13}$CH$_3$/$^{13}$CH$_3$; Phe-$^{13}$C$^{15}$N; Tyr-$^{13}$C$^{15}$N]-labeled RBD in complex with [U-$^2$H; Ala-$^{13}$CH$_3$; Met-$^{13}$CH$_3$; Ile-δ1-$^{13}$CH$_3$; Leu/Val-$^{13}$CH$_3$/$^{13}$CH$_3$]-labeled TF$^{ΔRBD}$. To corroborate intra-molecular distance restraints, 3D ($^1$H)-$^{13}$C HMQC-NOESY-$^1$H-$^{13}$C HMQC, 3D ($^1$H)-$^{15}$N HMQC-NOESY-$^1$H-$^{13}$C HMQC, 3D ($^1$H)-$^{13}$C HMQC-NOESY-$^1$H-$^{15}$N HMQC, $^{13}$C-edited NOESY-HMQC, $^{13}$C-edited NOESY-HSQC, $^{13}$C-edited HSQC-NOESY, $^{15}$N-edited NOESY-HMQC, and $^{15}$N-edited NOESY-HSQC were recorded on [U-$^2$H; Ala-$^{13}$CH$_3$; Met-$^{13}$CH$_3$; Ile-δ1-$^{13}$CH$_3$; Leu/Val-$^{13}$CH$_3$/$^{13}$CH$_3$; Phe-$^{13}$C$^{15}$N; Tyr-$^{13}$C$^{15}$N]-labeled RBD, [U-$^2$H; Ala-$^{13}$CH$_3$; Met-$^{13}$CH$_3$; Ile-δ1-$^{13}$CH$_3$; Leu/Val-$^{13}$CH$_3$/$^{13}$CH$_3$; Phe-$^{13}$C$^{15}$N; Tyr-$^{13}$C$^{15}$N]-labeled SBD, or $^{13}$C$^{15}$N-labeled PPD.

## Paramagnetic Relaxation Enhancement experiment

To observe paramagnetic relaxation enhancement (PRE), nitroxide spin label 1-oxyl-2,2,5,5-tetramethyl-3-pyrroline-3-methyl)-ethanethiosulfonate (MTSL, Toronto Research Chemicals Inc.) were introduced via cysteine-specific modification of TF K46C. Wild type TF has no cysteine residues. K46

mutant and its MTSL derivatives were determined not to perturb the TF structure, as assessed by $^1$H-$^{13}$C HMQC spectra. After purification, [U-$^2$H; Met-$^{13}$CH$_3$; Ile-δ1-$^{13}$CH$_3$; Leu/Val-$^{13}$CH$_3$/$^{13}$CH$_3$]-labeled TF K46C was exchanged into tris buffer (50 mM Tris-HCl pH 7.0, 50 mM KCl, and 1 mM β-mercaptoethanol). β-mercaptoethanol was removed by Zeba spin desalting column (Thermo Scientific, Waltham, MA) according to the manufacturer's protocol. MTSL was added from a concentrated stock in acetonitrile at a 10-fold excess, and the reaction was allowed to proceed at 4°C for ~12 hr. Excess MTSL was extensively removed by an Amicon stirred cell. PREs were observed from $^1$H-$^{13}$C HMQC spectra of TF in the absence and presence of PhoA by measuring peak intensities before (paramagnetic) and after (diamagnetic) reduction of the nitroxide spin label with ascorbic acid.

## Structure determination of TF dimer

The resonances of the full-length dimeric TF (~100 kDa) were assigned by a domain-parsing approach as reported previously (*Saio et al., 2014*). Near-complete assignment of TF was achieved for the resonances from methyl side chain, aromatic side chain, and amide group. The structure of TF dimer was calculated by CYANA 3.97 (*Güntert, 2004*) using the NOE-derived distance restraints, dihedral angle-restraints, and hydrogen bond restraints. PREs were solely used to monitor TF monomerization upon the addition of the substrate protein (*Figure 1—figure supplement 1A*), and were not used in the structure calculation. NOE peak lists were obtained from 3D ($^1$H)-$^{13}$C HMQC-NOESY-$^1$H-$^{13}$C HMQC, 3D ($^1$H)-$^{15}$N HMQC-NOESY-$^1$H-$^{13}$C HMQC, 3D ($^1$H)-$^{13}$C HMQC-NOESY-$^1$H-$^{15}$N HMQC, 3D-SOFAST-($^1$H)-$^{13}$C HMQC-NOESY-$^1$H-$^{13}$C HMQC and $^{13}$C-edited SOFAST-NOESY-HMQC, $^{13}$C-edited NOESY-HMQC, $^{13}$C-edited NOESY-HSQC, $^{13}$C-edited HSQC-NOESY, $^{15}$N-edited NOESY-HMQC, and $^{15}$N-edited NOESY-HSQC. Substantial number of inter- and intra-molecular NOEs were observed from NOESY spectra recorded on full length TF. The NOE restraints were further corroborated by the NOEs observed from isolated RBD in complex with TF$^{ΔRBD}$. The chemical shift perturbation profiles as well as NOEs observed for RBD-TF$^{ΔRBD}$ complex were consistent with those observed for full length TF, supporting the idea that the binding mode in the TF dimer is preserved in the interaction between the isolated domains. The intramolecular restraints obtained from NOESY experiments on full length TF were also corroborated by NOEs observed from the isolated domains of PPD, SBD and RBD. Note that most of the intramolecular NOEs from the isolated domains were consistent with the NOEs observed from TF dimer. A few intra-molecular NOEs observed from the isolated domains especially from the regions close to the dimer interface and the hinge regions were excluded in the calculation. Accordingly more than 2500 intramolecular NOEs as well as 54 intermolecular NOEs were collected for structure calculation (*Table 1*) (*Figure 2—figure supplement 4*). NOE restraints were corroborated by dihedral angle restraints derived from TALOS + (*Shen et al., 2009*) and hydrogen bond restraints added for the regions forming secondary structures as judged by the NOEs and TALOS+-derived dihedral angles. Intermolecular hydrogen bond restraints were added for the pair of atoms located close in the majority of the conformers in the NOE-derived preliminary structure. For the core region of RBD remote to the dimer interface, distance restraints from the crystal structure (*Ferbitz et al., 2004*) were loosely added to maintain overall fold of RBD. The 20 lowest-energy structures resulted from CYANA calculation were refined by restrained molecular dynamics in explicit water with CNS (*Brunger, 2007*). All of the intermolecular NOEs were well satisfied in the structure. The coordinates, restraints, chemical shift assignments have been deposited to PDB and BMRB.

## SEC-MALS experiments

Size-exclusion chromatography with multi-angle light scattering (SEC-MALS) was measured using DAWN HELEOS-II (Wyatt Technology Corporation) downstream of a Shimadzu liquid chromatography system connected to Superdex 200 10/300 GL (GE Healthcare) gel filtration column, or using DAWN HELEOS8+ (Wyatt Technology Corporation) downstream of TOSOH liquid chromatography system connected to TSKgel G3000SWXL (TOSOH Corporation) gel filtration column. In both instruments, the differential refractive index (Shimadzu Corporation) downstream of MALS was used to obtain protein concentration. The running buffer was 20 mM potassium phosphate (pH 7.0), 100 mM KCl, 4 mM β-mercaptoethanol, and 0.5 mM EDTA. 100 ~ 200 µL of the sample was injected with a flow rate of 0.5 ~ 1.0 mL min$^{-1}$. The data were analyzed with ASTRA version 6.0.5 or 7.0.1 (Wyatt Technology Corporation). To obtain the dissociation constant ($K_d$) of TF dimer, TF was

injected at varying concentrations, followed by $K_d$ estimation based on the weight-averaged molar mass as determined by SEC-MALS and protein concentration at the peak top, using the following equation.

$$M_w = M_m \left( \frac{8[M]_T + k_d - \sqrt{k_d^2 + 8[M]_T k_d}}{4[M]_T} \right)$$ (1)

where $M$w is the weight average molar mass obtained by SEC-MALS, $[M]_T$ is the molar concentration of protein (as measured by change in refractive index), and $M_m$ is molecular mass of the monomer. Nonlinear least square fitting was performed using Prism 5 (GraphPad Software).

## Analytical ultracentrifugation experiments

Sedimentation velocity experiments were conducted in a ProteomeLab XL-I analytical ultracentrifuge (Beckman Coulter, Indianapolis, IN) following standard protocols unless mentioned otherwise (*Benfield et al., 2011*). The samples, dialyzed overnight against the reference buffer (50 mM sodium phosphate pH 7.0, 100 mM NaCl) were loaded into a cell assembly comprised of a double sector charcoal-filled centerpiece with a 12 mm path length and sapphire windows. Buffer density and viscosity were determined in a DMA 5000 M density meter and an AMVn automated micro-viscometer (both Anton Paar, Graz, Austria), respectively. The partial specific volumes and the molecular masses of the proteins were calculated based on their amino acid compositions in SEDFIT (https://sedfitsedphat.nibib.nih.gov/software/default.aspx). The cell assembly, containing identical sample and reference buffer volumes of 360 µL, was placed in a rotor and temperature equilibrated at rest at 20°C for 2 hr before it was accelerated from 0 to 50,000 rpm. Absorbance scans at 230 and 280 nm were collected continuously for 12 hr. The velocity data were modeled with diffusion-deconvoluted sedimentation coefficient distributions c(s) in SEDFIT (https://sedfitsedphat.nibib.nih.gov/software/default.aspx), using algebraic noise decomposition and with signal-average frictional ratio and meniscus position refined with non-linear regression. The s-values were corrected for time and finite acceleration of the rotor was accounted for in the evaluation of Lamm equation solutions (*Benfield et al., 2011*). Maximum entropy regularization was applied at a confidence level of P-0.68.

Sedimentation velocity isotherm data that is the signal-weighted average sedimentation coefficients, sw(c), of the total sedimenting system derived from integration of the complete c(s) distributions at various concentrations (40.18, 12.13, 3.83 and 0.596 µM) of TF were fitted to a monomer-dimer self-association model using SEDPHAT (https://sedfitsedphat.nibib.nih.gov/software/default.aspx). For interacting systems, sw represents the average sedimentation property of the species under investigation. The association scheme used in this analysis was A + A ⟷ A2 with equilibrium dissociation constant $K_d$. All plots were generated with the program GUSSI (kindly provided by Dr. Chad Brautigam).

## Stopped-flow experiments

Kinetic measurements were performed on FP-8300 Fluorescence Stopped Flow System (JASCO Corporation). The excitation and emission wavelengths were set at 280 nm (band width 10 nm) and 350 nm (band width 20 nm), respectively, so that the intrinsic tryptophan-fluorescence of PhoA$^{220-310}$ containing two Trp residues or that of TF containing one tryptophan residue can be monitored. All measurements were carried out in the buffer containing 20 mM potassium phosphate (pH 7.0), 100 mM KCl, 4 mM β-mercaptoethanol, 0.5 mM EDTA, 0.05% NaN$_3$. Individual kinetics were typically measured 40 times and averaged. The data were analyzed with Prism 5 (GraphPad Software). To account for photobleaching, an exponential baseline was defined using the data after 1000 ms of the mixing, by which the dissociation or association has completed and reached to the equilibrium. Dissociation of TF dimer was initiated by 10-fold dilution of TF at 1 µM. The temperature was set to 22°C. The protein solution was placed in 2.5 mL syringe and the buffer was placed in 10 mL syringe. The dissociation kinetics was analyzed using a single exponential function. Binding between TF and PhoA$^{220-310}$ was monitored after rapid mixing by the stopped-flow instrument. Association of PhoA$^{220-310}$ and TF or TF$^{mon}$ was initiated by mixing equal volumes of 4 µM PhoA220-310 and 0–80 µM TF or TF$^{mon}$ resulting in final concentrations of 2 µM PhoA$^{220-310}$ and 0–40 µM TF or TF$^{mon}$. A single tryptophan residue in TF (W151) was mutated to phenylalanine in order to selectively monitor

the change in the fluorescence from PhoA$^{220-310}$ containing two residues both located in the binding sites for TF (*Saio et al., 2014*). Both samples were placed in the 10 mL syringe. The temperature was set to 18°C. The fluorescence intensity of PhoA$^{220-310}$ increased upon binding to TF as seen in the previous report using reduced and carboxymethylated form of α-lactalbumin (RCM-La). When PhoA$^{220-310}$ was mixed with the monomeric mutant TF$^{mon}$, each of the time traces was well explained by a single exponential curve. The time traces at varying concentration of TF$^{mon}$ showed linear dependence of the observed rate $k_{obs}$ on the concentration of TF$^{mon}$, and $k_{on}$ and $k_{off}$ were extracted by fit of the data to the linear function of $k_{obs} = k_{on}[TF]+k_{off}$. On the other hand, the time traces of the binding between PhoA$^{220-310}$ and TF were best represented as the sum of the two exponential curves. The fit of the time traces to two exponential functions resulted the fraction of the fast phase more than 80% that increased as the concentration of TF increased. The fraction for the fast phase coincides with the fraction of the dimer as estimated by the $K_d$ of dimerization (2 μM) determined by the AUC experiment, and thus we concluded that the fast and slow phases are attributed to the binding of PhoA$^{220-310}$ to the dimer and the monomer fractions of TF, respectively. The concentration for the plots of $k_{obs}$ was calculated for each of the dimer and the monomer, using the $K_d$ of dimerization (2 μM). The kinetic parameters determined for the monomer fraction of TF roughly correspond to those determined for the monomeric mutant TF$^{mon}$.

## ITC experiments

For the ribosome and TF, calorimetric titrations were carried out on iTC200 microcalorimeter (GE healthcare) at 22°C. All protein samples were dialyzed against ITC buffer containing 20 mM HEPES, pH 7.5, 50 mM potassium acetate, 20 mM MgCl$_2$, and 1 mM tris(2-carboxyethyl)phosphine (TCEP). The 200 μL sample cell was filled with 12 μM solution of the ribosome, and 40 μL injection syringe was filled with 160 to 190 μM solution of TF or RBD. The titrations were carried out with a preliminary 0.2 μL injection, followed by 14 injections of 2.5 μL each with time intervals of 5 min. The solution was stirred at 1000 rpm. For unfolded substrates (PhoA$^{220-310}$ or MBP$^{198-265}$) and TF, TF$^{mon}$, or TF$^{ΔRBD}$, calorimetric titrations were carried out on Auto-iTC200 microcalorimeter (GE healthcare). The calorimetric titrations for PhoA220-310 and MBP198-265 were performed at 8°C and 22°C, respectively. All protein samples were purified in ITC buffer containing 20 mM potassium phosphate (pH 7.0), 100 mM KCl by gel filtration. For titration of PhoA$^{220-310}$, the 200 μL sample cell was filled with 90 μM solution of TF, TF$^{mon}$, or TF$^{ΔRBD}$, and 40 μL injection syringe was filled with 1.1 mM solution of PhoA$^{220-310}$. For titration of MBP$^{198-265}$, the 200 μL sample cell was filled with 110 μM solution of TF or TF$^{ΔRBD}$, and 40 μL injection syringe was filled with 1.1 mM solution of MBP198-265. The titrations were carried out with a preliminary 0.2 μL injection, followed by 8 injections of 4.2 μL each with time intervals of 5 min. The solution was stirred at 1000 rpm. Data for the preliminary injection, which are affected by diffusion of the solution from and into the injection syringe during the initial equilibration period, were discarded. Binding isotherms were generated by plotting heats of reaction normalized by the modes of injectant versus the ratio of total injectant to total protein per injection. The data were fitted with Origin 7.0 (OriginLab Corporation, Northampton, MA).

## Anti-aggregation assays

Aggregation of denatured GAPDH from rabbit muscle (Sigma; G-2267) was measured as described previously (*Saio et al., 2014*). 125 μM GAPDH was denatured by 3 M guanidine-HCl in 20 mM potassium phosphate, pH 7.0, 100 mM KCl, 4 mM β-mercaptoethanol, 0.5 mM EDTA, and 0.05% NaN$_3$ for 12 hr at 4°C. The denatured GAPDH was diluted 50-fold into the buffer that does not contain guanidine-HCl and aggregation was monitored by 90° light scattering at 620 nm on a spectrofluorometer (FP-8500, JASCO Corporation) in the absence or presence of TF or TF$^{mon}$ at the concentration of 0.5 μM or 1 μM. The experiment was carried out at 20°C. The reproducibility was confirmed by independent assays repeated three times.

In anti-aggregation assay on OmpA$^{1-192}$, 62 μM OmpA$^{1-192}$ in 50 mM Tris-HCl pH 8.0, 500 mM NaCl, 400 mM imidazole, and 8 M urea was diluted 20-fold into 20 mM potassium phosphate, pH 7.0, 100 mM KCl, 4 mM β-mercaptoethanol, 0.5 mM EDTA, and 0.05% NaN$_3$. Aggregation was monitored by 90° light scattering at 620 nm on a spectrofluorometer (FP-8500, JASCO Corporation) in the absence or presence of TF or TF$^{mon}$ at the concentration of 4 μM. The experiment was carried out at 25°C.

## MBP refolding assay

Refolding experiments of the precursor form of MBP, preMBP, and slower folding mutant, $MBP^{Y283D}$, were performed as described before (*Huang et al., 2016*) with some modifications. The proteins were denatured in the buffer containing 100 mM HEPES, pH 7.5, 20 mM potassium acetate, 5 mM magnesium acetate, and 8 M urea. PreMBP and $MBP^{Y283D}$ were concentrated to 80 and 32 µM, respectively. Refolding of preMBP was initiated by 20-fold rapid dilution into the buffer containing 50 mM sodium phosphate, pH 7.0, 150 mM NaCl, and 0.05% $NaN_3$. Refolding process of preMBP in the absence and presence of TF or $TF^{\Delta RBD}$ at the concentration of 4 or 20 µM was monitored by an increase in tryptophan fluorescence intensity. Fluorescence intensity was measured using a microplate reader (Infinite 200 PRO, Tecan). The excitation and emission wavelengths were set at 295 nm (band width 5 nm) and 335 nm (band width 20 nm), respectively. The refolding was performed three times and averaged. All measurements were performed at 25°C. Data were analyzed by Prism 5 (GraphPad Software) using single exponential function. Refolding of $MBP^{Y283D}$ was initiated by 20-fold rapid dilution into the buffer containing 100 mM HEPES, pH 7.5, 20 mM potassium acetate, 5 mM magnesium acetate and the refolding process of $MBP^{Y283D}$ in the absence and presence of TF or $TF^{mon}$ at the concentration of 10 or 20 µM was monitored by an increase in tryptophan fluorescence intensity. Fluorescence intensity was measured using a spectrofluorometer (FP-8500, JASCO Corporation). The excitation and emission wavelengths were set at 295 nm (band width 2.5 nm) and 335 nm (band width 5 nm), respectively. The refolding was performed three times and averaged. All measurements were performed at 25°C.

## Acknowledgements

We thank P Rossi (St. Jude Children's Research Hospital), H Kumeta (Hokkaido University), Y Kumaki (Hokkaido University), Y Xia (St. Jude Children's Research Hospital), and S Kim (Rutgers University) for their help with setting up NMR experiments, K Maenaka (Hokkaido University) and Center for Research and Education on Drug Discovery for the use of iTC200, M Inouye (Robert Wood Johnson Medical School) for providing the RT expression plasmid. This work was supported by National Institutes of Health grant GM122462 (to CGK) and by JSPS KAKENHI (17H05657) and PRESTO JST (to TS). Some of the NMR experiments were performed at Hokkaido University Advanced NMR Facility, a member of NMR Platform.

## Additional information

### Funding

| Funder | Grant reference number | Author |
| --- | --- | --- |
| Japan Society for the Promotion of Science | KAKENHI (17H05657) | Tomohide Saio |
| Japan Science and Technology Agency | PRESTO | Tomohide Saio |
| National Institute of General Medical Sciences | GM122461 | Charalampos G Kalodimos |

The funders had no role in study design, data collection and interpretation, or the decision to submit the work for publication.

### Author contributions

Tomohide Saio, Conceptualization, Data curation, Formal analysis, Investigation, Methodology, Conception and design, Acquisition of data, Analysis and interpretation of data, Drafting and revising the article; Soichiro Kawagoe, Investigation, Acquisition of data, Analysis and interpretation of data; Koichiro Ishimori, Investigation, Analysis and interpretation of data, Revising the article; Charalampos G Kalodimos, Conceptualization, Supervision, Funding acquisition, Writing—original draft, Project administration, Writing—review and editing, Conception and design, Analysis and interpretation of data, Drafting and revising the article

## Author ORCIDs

Charalampos G Kalodimos (iD) https://orcid.org/0000-0001-6354-2796

## Decision letter and Author response

Decision letter https://doi.org/10.7554/eLife.35731.025
Author response https://doi.org/10.7554/eLife.35731.026

## Additional files

### Supplementary files

• Transparent reporting form
DOI: https://doi.org/10.7554/eLife.35731.020

### Data availability

Atomic coordinates for the TF dimer structure have been deposited in the Protein Data Bank (ID 6D6S).

The following dataset was generated:

| Author(s) | Year | Dataset title | Dataset URL | Database, license, and accessibility information |
|---|---|---|---|---|
| Tomohide Saio, Soichiro Kawagoe, Koichiro Ishimori, Charalampos G Kalodimos | 2018 | Solution structure of Trigger Factor dimer | http://www.rcsb.org/pdb/search/structid-Search.do?structureId=6D6S | Publicly available at the RCSB Protein Data Bank (accession no. 6D6S) |

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
