## [Decision Letter]

Thank you for submitting your article "Oligomerization of a molecular chaperone modulates its activity" for consideration by *eLife*. Your article has been favorably evaluated by John Kuriyan (Senior Editor) and three reviewers, one of whom, Lewis E Kay (Reviewer #1), is a member of our Board of Reviewing Editors.

The reviewers have discussed the reviews with one another and the Reviewing Editor has drafted this decision to help you prepare a revised submission.

Your work has been reviewed by three experts who are all very positive but suggest minor improvements for a subsequent revision. In particular, all reviewers noted that the structure of the TF dimer is a major accomplishment both technically and in terms of the information that it provides concerning mechanism of action. Indeed, reviewers write: "This work makes important contributions to the chaperone biology by rationalizing observations in structural terms. Moreover, the structural studies of such a large complex are far from routine and the work thus serves as an excellent example of the power of modern NMR approaches. Overall this is an excellent paper". "This is a major contribution because:

- TF is representative of ATP-independent chaperones, which are not as well understood as ATP-dependent chaperones, such as GroEL.

- The authors were able to cleverly overcome major technical challenges posed by the high MW and the dynamic nature of the monomer/dimer exchange.

- The monomer/dimer equilibrium is central to the chaperone function of TF, not only as a modulatory mechanism, but also because it is known that TF binds, as a monomer, at the ribosome to prevent the aggregation of nascent polypeptides, while it functions as a dimer in cytosol to assist protein folding and prevent aggregation of unfolded protein.

- The structure informs a new molecular mechanism for the TF chaperone (Figure 5)"

"The widely accepted concept is that TF dimerization is a storage form of the chaperone. An alternative function of the TF dimer in binding folded, small substrates was suggested by the crystal structure of the Thermotoga maritima TF in complex with the ribosomal protein S7 (Martinez-Hackert and Hendrickson, 2009). Saio et al. elegantly rule out the second scenario, by showing that it is the monomeric TF that binds two S7 molecules and that the S7 protein is unfolded when bound."

We would encourage the authors to address point 6 below experimentally and perhaps aspects of point 1, but acceptance is not predicated on new experimental data.

Major:

1) The authors utilize a mutant TF, ensuring that the kinetic parameters obtained refer to the monomeric form of the protein. But I wonder about the parameters for the dimer. Since substrate can bind to monomers with higher affinity than dimers (as the authors mention) and dimer dissociation is a route for substrate binding, are the measured on/off rates for dimer contaminated by the monomer-dimer equilibrium (even though the equilibrium amount of dimer is much higher than monomer)? Is it possible to prepare a disulfide-linked dimer that would provide a better mimic of the dimeric chaperone, without complications? The question is of interest as the authors argue that the better anti-aggregation activity of the dimer reflects in part the fact that the dimer can dissociate to form 2 monomers, which are then both available for binding. If the dimer could not dissociate into monomers then would the monomer TF be a more potent anti-aggregator or would the faster on rate for the dimer still play an effect? Given that the KDs for dimerization, and for substrate binding for both monomer and dimeric TF are available and assuming physiological concentrations of TF (50 μM) and of substrate it should be possible to calculate the fraction of substrate that is bound to TF that is a monomer or a dimer.

2) In Figure 3 the authors speak about a holdase activity for the dimeric TF form. They rationalize this in terms of the faster association rates measured. Yet, naïvely I would have thought that what is important for a holdase activity is the lifetime of the bound complex or 1/*k_off_*; and *k_off_*is much larger for the dimer than the monomeric TF. (ii) Why is there not a holdase effect on the WT MBP protein? (iii) I would imagine that for TF to prevent aggregation the substrate on rate must be greater than the aggregation rates. Can the authors compare their measured on rates to aggregations rates to verify this if aggregation rates are available?

3) The spectrum of the dimeric TF shows significant broadening of residues from the RBD, seen both in cases of amide resonances (Morgado et al., 2017) and of methyl group selective labeling (Saio et al., 2014). A major concern is that the current structure does not explain that broadening, that most likely reports on a dynamic assembly. This then raises the question – how well does this single conformation satisfy all the distance restraints?

Also, due to the broadening of the RBD residues, how many NOEs were reliably measured between the RBD and the other TF domain in the dimer? Also it is important that the chemical shifts are deposited in the BMRB, and that the scripts containing restraints used for structure calculations be including in supporting information.

4) The observation that the wild type "dimeric" TF has better "chaperone" activity than the monomeric TF is a bit confusing. As TF monomerizes upon binding to substrate the overall aggregation activity should in principle be quite similar between the wild type and mutant TF.

5) Another point is that the chaperone activities of wild type TF and TF^ΔRBD^ truncation that results in TF monomerization were found to be very similar (Merz et al., 2006). Perhaps the authors can also compare between wt ΔRBD monomer TF for their substrates

6) Last point regarding the chaperone activity is that the prevention of GAPDH aggregation was done using 0.5μm of TF dimer which is significantly below the K_D_ for dimer formation. It would be beneficial to repeat the experiment using higher concentrations of TF (but same GAPDH: TF ratio) to see whether the effect is bigger for fully formed TF dimers.

---

## [Author Response]

Major:1) The authors utilize a mutant TF, ensuring that the kinetic parameters obtained refer to the monomeric form of the protein. But I wonder about the parameters for the dimer. Since substrate can bind to monomers with higher affinity than dimers (as the authors mention) and dimer dissociation is a route for substrate binding, are the measured on/off rates for dimer contaminated by the monomer-dimer equilibrium (even though the equilibrium amount of dimer is much higher than monomer)? Is it possible to prepare a disulfide-linked dimer that would provide a better mimic of the dimeric chaperone, without complications? The question is of interest as the authors argue that the better anti-aggregation activity of the dimer reflects in part the fact that the dimer can dissociate to form 2 monomers, which are then both available for binding. If the dimer could not dissociate into monomers then would the monomer TF be a more potent anti-aggregator or would the faster on rate for the dimer still play an effect? Given that the KDs for dimerization, and for substrate binding for both monomer and dimeric TF are available and assuming physiological concentrations of TF (50 μM) and of substrate it should be possible to calculate the fraction of substrate that is bound to TF that is a monomer or a dimer.

We had previously tried to prepare a disulfide-linked TF dimer but unfortunately it proved not to be a good mimic of the physiological dimer (and thus we decided not to include the data in the original submission). We designed a mutant TF (I80C/E383C) in which the two surface amino acid residues located at the RBD-Arm2 interface in the dimer were mutated to Cys. Based on the structural data this pair was expected to form a favorable disulfide bond. SEC-MALS showed that the oxidized form of TF I80C/E383C mutant indeed forms a stable dimer, even at concentrations below the dimerization K_D_ of (~2 μM). NMR spectra of the oxidized TF I80C/E383C showed that the chemical shift of most of the resonances matched those of wt TF. However, several resonances corresponding to residues located at the RBD-Arm2 interface were broadened in the spectrum of the mutant, which suggests that the S-S bridge may have destabilized the local structure around the substrate-binding site B and/or alter the dynamics of the dimeric interface. The stopped-flow experiments indeed showed that the association kinetics between the TF I80C/E383C mutant and protein substrate is much slower than that between wt TF and protein substrate. Because the substrate-binding site-B on TF partially overlaps with the dimer interface, these results imply that the introduction of the intermolecular disulfide bridge or cysteine mutations disturbs the substrate binding site on TF. Since all the dimer interfaces partially overlap with substrate-binding sites, it has been impossible to introduce a disulfide bridge without disrupting substrate binding to TF.

2) In Figure 3 the authors speak about a holdase activity for the dimeric TF form. They rationalize this in terms of the faster association rates measured. Yet, naïvely I would have thought that what is important for a holdase activity is the lifetime of the bound complex or 1/k_off_; and k_off_ is much larger for the dimer than the monomeric TF. (ii) Why is there not a holdase effect on the WT MBP protein? (iii) I would imagine that for TF to prevent aggregation the substrate on rate must be greater than the aggregation rates. Can the authors compare their measured on rates to aggregations rates to verify this if aggregation rates are available?

As shown in our previous work by Huang et al., 2016, the kinetic competition between the substrate folding rate (*k*_fold_) and its association rate with the chaperone (*k_on_*) determines the effect of the chaperone on the folding of the substrate protein. Although *k_off_*may also play a role, it is in this case of almost no relevance given that the dissociation rates between chaperones and substrates are relatively fast. Whether a chaperone delays the folding of a substrate is determined by how quickly it can associate with the unfolded state of the substrate before it has the chance to fold to its native state. We added relevant text elaborating this point in the eighth paragraph of the Results.

ii) The reason is that wt MBP has a fast folding rate and thus TF has no effect. Even strong holdase chaperones such as SecB have little effect on the folding rate of wt MBP. More details and kinetic rates are provided in our previous work (Huang et al., 2016).

(iii) Assuming that the binding kinetics for the unfolded GAPDH is similar to that for the unfolded PhoA, the association rate between TF and GAPDH is expected to be ~0.5 s^-1^, which is indeed much faster than the process of overall aggregation (Figure 4A). There are no rates for the individual processes leading to aggregation and thus a direct comparison is currently not possible.

3) The spectrum of the dimeric TF shows significant broadening of residues from the RBD, seen both in cases of amide resonances (Morgado et al., 2017) and of methyl group selective labeling (Saio et al., 2014). A major concern is that the current structure does not explain that broadening, that most likely reports on a dynamic assembly. This then raises the question – how well does this single conformation satisfy all the distance restraints?Also, due to the broadening of the RBD residues, how many NOEs were reliably measured between the RBD and the other TF domain in the dimer? Also it is important that the chemical shifts are deposited in the BMRB, and that the scripts containing restraints used for structure calculations be including in supporting information.

The measured kinetics of TF dimer association and dissociation suggest that the broadening of resonances corresponding to residues located at the dimeric interface is due to the monomer-dimer transition, rather than internal dynamics within TF. This is further corroborated by the observation that the methyl TROSY spectra of dimeric TF at higher concentrations exhibit reduced line broadening. The spectra reported in the current work are of much higher quality than the ones reported in Saio et al., 2014 as a result of optimized experimental conditions (Figure 2—figure supplement 1). We have added a in the subsection “Structure of dimeric TF”. All of the intermolecular NOEs are well satisfied in the structure as shown in Author response image 1, indicating that the dimeric TF structure is faithfully represented by our NOE-derived distance restraints. All of the coordinates, restraints, chemical shift assignments are being deposited to PDB and BMRB. We have added relevant text to the subsection “Structure determination of TF dimer”.

**Author response image 1. respfig1:** NOE mapping for the interfaces for PPD-RBD and Arm1-RBD (left panel) and Arm2-RBD. The intermolecular NOEs are indicated by red lines. TF domains are colored as in Figure 2.

4) The observation that the wild type "dimeric" TF has better "chaperone" activity than the monomeric TF is a bit confusing. As TF monomerizes upon binding to substrate the overall aggregation activity should in principle be quite similar between the wild type and mutant TF.

We propose here that the difference in the chaperone activities between the dimeric TF and the monomeric TF is due to the differences in the binding kinetics to the substrate protein and the local concentration. Especially for anti-aggregation activity, the elevated local concentration of TF by dimerization is a major factor as explained in the Discussion. The binding to the substrate leads to the dissociation of the dimer into monomers as shown by SEC-MALS (Figure 1—figure supplement 1E) and NMR (Figure 1—figure supplement 1A). Upon dissociation, one dimer generates two monomeric TF that both co-localize nearby the substrate protein. For a long substrate protein consisting of > ~200 amino acid residues such as GAPDH (~350 a.a.) and MBP (~400 a.a.), both of TF monomers are used to capture the substrate protein (Figure 6, panels vii and viii). Thus, the dissociation of TF dimer into monomers upon binding to the substrate protein results in higher local concentration of TF around the substrate protein, resulting in faster association of the second TF molecule to the substrate protein. The faster association enables TF to more efficiently capture the substrate protein before the substrate proteins aggregate. Such an effect is not observed when a shorter substrate is used, in which case only one TF molecule is sufficient to bind the entire substrate. We have added relevant text to the Discussion.

5) Another point is that the chaperone activities of wild type TF and TF^ΔRBD^ truncation that results in TF monomerization were found to be very similar (Merz et al., 2006). Perhaps the authors can also compare between wt ΔRBD monomer TF for their substrates

Merz et al. tested several TF variants including TF wt and TF^ΔRBD^ and showed the decreased anti-aggregation activity as well as decreased holdase activity for TF^ΔRBD^. These observations are consistent with our results. In our GAPDH aggregation assay, as well as MBP refolding assay, the effect of TF^ΔRBD^ was also assessed and the results showed that the activity of TF^ΔRBD^ was essentially the same as that of the TF monomeric mutant. We added relevant text about TF^ΔRBD^ and cited the paper in the first paragraph of the subsection “Dimerization modulates the chaperone activities of TF”.

6) Last point regarding the chaperone activity is that the prevention of GAPDH aggregation was done using 0.5 μm of TF dimer, which is significantly below the K_D_ for dimer formation. It would be beneficial to repeat the experiment using higher concentrations of TF (but same GAPDH: TF ratio) to see whether the effect is bigger for fully formed TF dimers.

We also performed GAPDH aggregation assay in the presence of TF variants at higher concentrations. At TF concentration of 1 μM, the monomeric mutant (TF^mon^) exhibited weaker anti-aggregation activity than wt TF, which is consistent with the result in the presence of 0.5 μM of TF and TF^mon^. However, the difference between dimeric TF and monomeric TF is smaller at 1 μM concentration because at this higher concentration even the monomeric TF can efficiently suppress aggregation. We have added a new figure (Figure 4—figure supplement 1). We did not increase the concentration of GAPDH because the aggregation is very sensitive to the experimental conditions including concentration of protein and denaturant, dilution factor, and temperature, and thus change of GAPDH concentration would require extensive optimization. Almost all previous GAPDH aggregation assays have been performed using the similar conditions to the ones used in this work.